# Efficient and Reusable Sorbents Based on Nanostructured BN Coatings for Water Treatment from Antibiotics

**DOI:** 10.3390/ijms232416097

**Published:** 2022-12-17

**Authors:** Kristina Yu. Kotyakova, Liubov Yu. Antipina, Pavel B. Sorokin, Dmitry V. Shtansky

**Affiliations:** Laboratory of Inorganic Nanomaterials, National University of Science and Technology “MISIS”, Leninsky Prospect 4, 119049 Moscow, Russia

**Keywords:** BN coatings, antibiotics, wastewater treatment, sorption capacity, DFT calculations

## Abstract

Increasing contamination of wastewater with antibiotics used in agriculture, animal husbandry, and medicine is a serious problem for all living things. To address this important issue, we have developed an efficient platform based on a high specific surface area hexagonal boron nitride (BN) coating formed by numerous nanopetals and nanoneedles. The maximum sorption capacity of 1 × 1 cm^2^ BN coatings is 502.78 µg/g (tetracycline, TET), 315.75 µg/g (ciprofloxacin, CIP), 400.17 µg/g (amoxicillin, AMOX), and 269.7 µg/g (amphotericin B, AMP), which exceeds the sorption capacity of many known materials. Unlike nanoparticles, BN-coated Si wafers are easy to place in and remove from antibiotic-contaminated aqueous solutions, and are easy to clean. When reusing the adsorbents, 100% efficiency was observed at the same time intervals as in the first cleaning cycle: 7 days (TET) and 14 days (CIP, AMOX, AMP) at 10 µg/mL, 14 days (TET, CIP, and AMOX) and 28 days (AMP) at 50 µg/mL, and 14 days (TET) and 28 days (CIP, AMOX and AMP) at 100 µg/mL. The results obtained showed that TET and CIP are best adsorbed on the surface of BN, so TET was chosen as an example for further theoretical modeling of the sorption process. It was found that adsorption is the main mechanism, and this process is spontaneous and endothermic. This highlights the importance of a high specific surface area for the efficient removal of antibiotics from aqueous solutions.

## 1. Introduction

The era of antibiotics began in the early 1940s, shortly after the discovery of penicillin by Alexander Fleming in 1928. With population growth and urbanization, the need for antibiotics has steadily increased. Antibiotic use, according to a World Health Organization report, increased by 91% worldwide and 165% in low-income countries from 2000 to 2015 [1], and continues to grow. The global market for antibiotics is projected to grow from $38.08 billion in 2021 to $45.30 billion in 2028 [2]. As a result, antibiotic contamination of wastewater is becoming a significant public health problem [3,4,5]. The presence of antibiotic residues in water has detrimental effects on humans, animals, and other creatures because it contains carcinogenic and toxic compounds. About 70% of antibiotics administered to humans are excreted in a non-metabolized form as active compounds and end up in wastewater [6]. In addition, extremely high concentrations of antibiotics have been reported in wastewater from antibiotic production plants (oxytetracycline: 32.0 mg/L), livestock farms (oxytetracycline: 2.1 mg/L), hospitals (ciprofloxacin (CIP): 0.9 mg/L), urban environments (CIP: 0.25 mg/L), and near aquaculture farms (sulfamethoxazole: 5.6 mg/L) [7,8,9,10]. Although reported levels of their presence in the environment are generally low, ranging from ng/L to µg/L, antibiotics are “pseudo-stable” contaminants due to their constant release and presence in the environment [11]. Studies have identified the potentially toxic effects of hospital effluents entering the aquatic environment [12], and also found drug-resistant bacteria in areas where hospital effluents are present [13]. In particular, resistance rates to ciprofloxacin, commonly used to treat urinary tract infections, ranged from 8.4% to 92.9% for *Escherichia coli* (*E. coli*) and from 4.1% to 79.4% for *Klebsiella pneumoniae* (*K. pneumoniae*), respectively [14]. This leads to a decrease in the efficiency of antibiotic administration [15,16]. All this indicates the importance of developing effective approaches to the removal of antibiotics from the aquatic environment. Amoxicillin, ciprofloxacin, and tetracycline are among the classes of most commonly prescribed antibiotics [17,18,19,20]. Tetracycline and ciprofloxacin are major wastewater pollutants in East and Southeast Asia [21]. According to the World Health Organization report for 2015–2016, amoxicillin is one of the most used antibiotics in the world [22] and, therefore, is often detected in wastewater [23,24]. Amphotericin B is a common anti-fungal agent that has been used for more than half a century [25,26,27], traces of which are also often found in the environment [28]. Several microorganisms, such as *Aspergillus terreus*, were reported to develop resistance to amphotericin B [29], which requires its removal from wastewater.

For the treatment of wastewater containing pharmaceutical compounds, various physical, chemical, and biological processes, as well as their combinations, are used [30,31,32] but not all of them meet modern requirements for efficiency and cost. These methods have a number of disadvantages, such as introduction of active organisms into the aquatic environment during water treatment, low selectivity [33,34,35], rapid annihilation of photo-generated carriers, and incomplete mineralization [32]. Among the existing treatment methods, adsorption is one of the most promising and economical methods for removing pharmaceutical residues from wastewater [36,37,38,39]. Moreover, unlike other methods such as decomposition or chemical oxidation, this method does not produce secondary pollutants during purification [31,40].

Nanomaterials with a high specific surface area are a promising platform for inexpensive and highly effective sorbents of various polluting molecules [41,42]. Various materials have been studied for the adsorption of antibiotics from aqueous solutions: carbon nanotubes [43], graphene [44,45,46], graphene oxide [47], activated carbon [48], metal-organic framework [49,50,51], boron nitride [36,52], and others. 2D materials have a large specific surface area and therefore are excellent sorbents for a wide variety of pollutants. 

Hexagonal boron nitride (h-BN) stands out for its unique properties (non-toxicity, enhanced thermal stability, recyclability, superior oxidation resistance, high specific surface area, and chemical inertness) which make it a good adsorbent [53,54,55,56]. In addition, considering the possibility of large-scale synthesis of atomically thin h-BN nano-sheets (BNNSs), this material was studied as an adsorbent in water purification [36,56,57]. Since BN sorbents are very light and have a highly developed surface, they exhibit high gravimetric capacity, while their high chemical and thermal stability ensures good material regeneration. Hexagonal BN is a good sorbent even in a 2D form [58] (the maximum adsorption capacity of tetracycline is 346,66 mg/g), but modification of BN surface can increase its sorption capacity up to 500 mg/g (depending on modification) [59,60,61]. All this confirms that the rational modification and proper design of the h-BN material can provide excellent sorption characteristics of h-BN-based nanomaterial. However, BN is often used in powder form [58,60], which is not always convenient for industrial applications. The usage of h-BN coatings deposited on a substrate could bring us closer to a more practical material. Despite significant progress in wastewater treatment from antibiotics, the relevance of developing highly effective and safe adsorbents remains high. The development of adsorbents based on hexagonal boron nitride will make it possible to obtain an affordable and cost-effective reusable adsorbent for more efficient water purification from antibiotics. In addition, sorption to amphotericin B was studied for the first time.

Here, we investigated nano-structured h-BN-based coatings with a highly developed surface as a sorbent for various most commonly used antibiotics: ciprofloxacin (CIP), tetracycline (TET), amoxicillin (AMOX) and amphotericin B (AMP). We characterized the surface of the obtained BN-materials by various methods and studied their sorption capacity depending on the initial antibiotic concentration and solution pH both by experimental methods and by theoretical modeling. It is shown that after coating purification their sorption capacity decreases insignificantly, which makes it possible to reuse this material.

## 2. Results and Discussion

### 2.1. Characterization of BN Coatings

SEM micrographs of the as-synthesized BN coatings are shown in Figure 1a. The coating consists of spherical BN nanoparticles (NPs) formed by numerous nanosheets and nanoneedles. The size of BNNPs (Figure 1b) ranges from 90 to 600 nm, while the majority of NPs (>65%) are 200 to 400 nm in size. According to the EDX spectroscopy analysis, the main coating components are B (51.3%) and N (45.4%), although traces of O and C (3.3%) are also present. Figure 1c shows the FTIR spectrum of BN coating. The observed two high-intensity peaks can be attributed to out-of-plane B-N-B bending (780 cm^−1^) and in-plane B-N stretching (1370 cm^−1^) vibrations [62]. A small peak at 520 cm^−1^ corresponds to B-O bonds [63]. The XRD pattern of BN coating is presented in Figure 1d. Besides the main peaks from the (002), (100) and (101) BN crystallographic planes (ICDD card No. 00-034-0421), there are additional maxima corresponding to BNO (ICDD card No. 00-37-1234) and B_2_O_3_ (ICDD card No. 00-06-0297) phases.

The specific surface area of BN coating was measured by low-temperature nitrogen adsorption on a NOVA 1200e instrument (Quantachrome, Boynton Beach, FL, USA). The obtained results were processed using the Brunauer-Emmett-Teller (BET) equation. Prior to adsorption measurements, the samples were degassed in a vacuum at 200 °C overnight. The surface area of the nano-structured BN coating was 90.61 m^2^/g.

The zeta potential of BN was determined using a Zetasizer Nano-ZS ZEN3600 instrument (Malvern). The charge of the synthesized pure BN coating at pH 7 is −26 mV. It was reported that a change in the acidity of the medium did not affect the BN surface charge [64]; therefore, the BN charge was considered unchanged.

### 2.2. Kinetics of Antibiotic Adsorption on BN Coatings

Kinetic curves showing the removal efficiency of four types of antibiotics at various pH and initial antibiotic concentrations are presented in Figure 2. The purification efficiency depends on the treatment time. At initial antibiotic concentrations of 10, 50, and 100 µg/mL and pH 4, 100% efficacy is observed for TET on days 12, 15, and 18, for CIP on days 14, 21, and 23, for AMOX on days 17, 28 and 26, and AMP on days 23, 28, and 28, respectively. At the same initial concentrations and pH 7, 100% efficacy is achieved after 7, 9, and 10 days (TET), after 8, 10, and 11 days (CIP), after 10, 11, and 12 days (AMOX), and after 11, 12, and 14 days (AMP). In an alkaline environment (pH 9), the time required for complete solution purification from antibiotics is 11, 12, and 14 days for TET, 12, 14, and 14 days for CIP, 14, 14, and 15 days AMOX, and 18, 18, and 21 for AMP.

Antibiotic purification efficiencies (R_50_ and R_100_) using BN coatings are presented in Table 1. The purification efficiency depends on the pH of the medium and the type of antibiotic. The R_50_ values increase with the increase in the initial concentration of antibiotics. In an acidic environment (pH 4), R_50_ is reached after 2, 3, and 4 days (TET), 3, 4, and 4 days (CIP), 5, 5, and 8 days (AMOX) and 5, 6, and 8 days (AMP), respectively, at initial concentrations of 10, 50, and 100 µg/mL. In a neutral environment (pH 7), purification is faster. At initial concentrations of 10, 50, and 100 µg/mL, the R_50_ values are 2, 2, and 2 days (TET), 2, 3, and 3 days (CIP), 3, 3, and 4 days (AMOX), and 3, 4, and 5 days (AMP), respectively. In an alkaline environment (pH 9), the time to reach R_50_ increases again: 2, 2, and 3 days (TET), 3, 3, and 4 days (CIP), 4, 5, and 6 days (AMOX) and 5, 7, and 6 (AMP) days. The hydrogen index of wastewater from medical institutions into the sewer is 6.7–7.7 [51]. Thus, the obtained results show high prospects for the use of BN sorbents for wastewater treatment with pH~7 from antibiotics. The efficiency removal can be represented in a row: TET > CIP > AMOX > AMP.

The pH of a solution is one of the most important parameters that determine the effectiveness of the interaction between the adsorbate and the adsorbent. A change in pH affects not only the surface charge of the adsorbent, but also the degree of adsorbate ionization [51]. The acidity of the medium also significantly affects the adsorption kinetics. The TET molecule in an aqueous solution can enter into a protonation-deprotonation reaction with an increase in the solution pH [52]. In our case, with an increase in pH from 4 to 7, the adsorption properties increase, but in an alkaline medium, they decrease again. This can be explained by the fact that the TET molecule gradually becomes neutral or negatively charged because of the deprotonation reaction, which reduces its electrostatic interaction with the negatively charged BN surface.

In the case of CIP, an increase in adsorption properties is observed with an increase in pH from 4 to 7, which indicates a cation exchange adsorption mechanism. The subsequent decrease in the adsorption rate with an increase in pH to 9 may be due to the presence of the negative form of the CIP molecule, which leads to CIP repulsion from the negative BN surface. The observed dependence is similar to the CIP adsorption on the synthesized birnessite, where antibiotic adsorption first increased with increasing pH to 8.7, and then decreased [53]. 

AMOX adsorption efficiency increases with an increase in solution pH from 4 to 7, which may be due to an increase in the intensity of protonation of AMOX carbonyl groups and, as a result, an increase in the electrostatic interaction with the BN surface. A decrease in the AMOX removal rate with a further increase in pH to 9 may be because of the electrostatic repulsion of AMOX molecules by the BN surface, since both carboxyl and amino groups were deprotonated under alkaline conditions, which make AMOX negatively charged. Similar results were observed when studying the effect of pH change on the adsorption of AMOX trihydrate using activated charcoal from *Maerua Decumbens*: the adsorption efficiency first increased with increasing pH from 2 to 8, and then gradually decreased [54].

The AMP charge depends on pH and leads to different adsorption activity. In our case, with an increase in pH from 4 to 7, the adsorption properties increase, but in an alkaline medium they decrease again. This can be explained by the fact that the zeta potential increases from −34.5 to −26 mV with an increase in pH from 4 to 7, this is due to the dissociation of the carboxylate moiety of AMP [65]. A decrease in the rate of AMP removal with a further increase in pH to 9 may be because of electrostatic repulsion from the BN surface, since both the carboxyl and amino groups are deprotonated under alkaline conditions, which leads to more negatively charged AMP.

### 2.3. Cleaning BN Coatings from Adsorbed Antibiotics 

Figure 3 shows the FTIR spectra of the BN samples before and after cleaning from antibiotics at a maximum concentration of 100 µg/mL. The observed peaks of high intensity are related to vibrations of the BN bonds (B-N-B at 780 cm^−1^ and B-N at 1370 cm^−1^) [62]. Successful antibiotic adsorptions on the BN surface is confirmed by the presence of peaks of the functional groups included in their composition. After the CIP adsorption, characteristic peaks are observed in the FTIR spectrum at 1050–1000 cm^−1^ (C–F), 1270 cm^−1^ (C–N), 1624 cm^−1^ (C=C), 1725–1705 cm^−1^ (C=O) and 3150–2750 cm^−1^ (C–H ). A wide maximum in the range of 3700–3000 cm^−1^ is attributed to vibrations of the N–H and O–H bonds [66]. After the adsorption of TET, characteristic peaks are observed in the FTIR spectrum associated with stretching vibrations of the N–H and O–H bonds (3342–3325 cm^−1^), CH (3064–3003 cm^−1^) and CH_3_ (methyl) (2955–2835 cm^−1^), C=C bond (1622–1569 cm^−1^), bending vibrations of C–H (1454 cm^−1^) and CH3 (1357 cm^−1^), C–H bond (1247–1000 cm^−1^) and stretching vibrations of C–N bond (995 cm^−1^) [67]. The adsorption of AMOX on the BN surface is confirmed by the presence of –COOH and –NH_2_ groups (stretching vibrations of O–H bonds at 3400 cm^−1^ and –NH at 3166 cm^−1^). There is also a peak at 1034 cm^−1^ from in-plane deformation vibrations of C–H and N–H bonds. The peak at 1680 cm^−1^ can be attributed to vibrations of the C–O bond [68]. After adsorption of AMP, stretching vibrations of the C–O bond, bending vibrations of the C=O bond, and stretching vibrations of the C–H bond at 1020 cm^−1^, 1700 cm^−1^ and 2900 cm^−1^, respectively are observed [69].

Only after surface cleaning from adsorbed AMOX, low-intensity peaks are observed corresponding to C–O and O–H vibrations at 1680 and 3300 cm^−1^, respectively. In the IR spectra of samples after the removal of CIP, TET and AMP, only peaks corresponding to h-BN are seen. This indicates successful cleaning of the coating surface from antibiotics before reuse. 

The efficiency of antibiotic removal during coating reuse can be judged from the kinetic curves shown in Figure 4. The removal efficiency also increases with increasing contact time; however, the removal rate is lower than in the first cycle. Data on the efficiency of the primary and reuse of samples coated with BN at pH 7 are presented in Table 2. It can be seen that, except for AMP, at a concentration of 50 μg/mL, R_100_ is achieved over the same period. If we compare the R_100_ and R_50_ values on the 7th day of purification in more detail, then the repeated use of coatings reduces the efficiency by 0.0–10.9% (TET), 1.5–13.9% (CIP), 5.1–19.4% (AMOX), and 9.7–25.7% (AMP). It should be noted that there is no pronounced dependence of R on the initial concentration of the antibiotic when samples are reused.

TET and CIP adsorb on the surface of BN best of all and approximately equally. Therefore, TET was chosen as an example for further theoretical modeling of the sorption process. We considered TET in three states corresponding to three pH solutions: neutral (TET^0^), negatively (TET^−1^), and positively (TET^+1^) charged (Figure 5). It should be noted that, during optimization, the neutrally charged form rearranges into a zwitterion, in which a proton from the OH group is in a superposition between oxygen and two nitrogen atoms of amide and dimethylamine groups. The formation of charged forms occurs by removing or adding a proton to these groups. The interaction of all forms of TET on an ideal BN surface is considered. 

To assess which sorption process occurs, adsorption or absorption, a model of multilayer BN was built, on which an antibiotic molecule was placed (Figure 5b). The adsorption process was modeled by placing a TET molecule in the most energetically favorable configuration on the BN surface (Figure 5b, right). When simulating the absorption process, an antibiotic molecule was placed between the BN layers (Figure 5b, left).

First, in order to assess which sorption process occurs, adsorption or absorption, a model of multilayer boron nitride was built on which the antibiotic molecule was placed (Figure 5b). The adsorption process was modeled by placing tetracycline in the most energetically favorable configuration on the surface of the boron nitride layers (Figure 5b, right). The absorption process represented the arrangement of the antibiotic between the layers (Figure 5b, left).

It can be seen that the most stable vertical stacking of the TET molecule leads to a change in the geometry of the BN layers with the formation of a concavity on its surface, increasing the interaction area. Such a curvature disrupts the BN electronic structure and leads to a redistribution of the electron density between the antibiotic and h-BN. In the region of the oxygen-containing groups of the antibiotic, there is a rather strong transfer of electron density to boron atoms. This redistribution of electron density leads to the binding of the antibiotic and h-BN. However, the absorption of sufficiently large antibiotic molecules between the h-BN layers leads to the destruction of the van der Waals interaction between them, which significantly increases the substrate energy (by 2.5 eV). As a result, the adsorbed location of the antibiotic on the BN surface (Figure 5b, right) is 1.55 eV more favorable than that absorbed one between the layers (Figure 5b, left).

Figure 6 shows the dependence of the binding energy of TET to the BN surface as a function of the distance between them and the electron density redistribution for the three charged structures. Figure 6a shows that the antibiotic adsorption process on the BN surface is barrier-less with a binding energy of approximately 1.4–1.6 eV. The negatively charged form of TET, characteristic of pH > 7, exhibits a lower binding energy and a higher desorption barrier than the positively charged form, in good agreement with the experiment. As follows from Figure 6a, the dependence on the molecule charge is quite small and the sorption process is equally effective regardless of the medium pH with a slight predominance of neutral and negatively charged forms, which are characteristics of neutral and alkaline media. 

It follows from the observed charge redistribution (Figure 6b–d) that for neutral and negatively charged structures, electron density transfer occurs on the OH groups of the peripheral region, which handle the interaction of TET with BN. In the case of the negatively charged form (TET^−1^, Figure 6c), there is a region of strong redistribution on the amide group, which pulls the electron density from the BN to itself. In the case of a neutrally charged molecule (Figure 6b), it is difficult to identify the dominant interaction region; there is a uniform redistribution of electron density over the entire molecule. Such a uniform electron transfer ensures good antibiotic binding during its adsorption on the BN surface. In the case of TET^+1^ (Figure 6d), which corresponds to an acidic medium, the electron density redistribution is concentrated mainly on the antibiotic molecule and almost does not affect BN, which explains the weaker binding compared to TET^−1^. 

Table 3 shows the maximum adsorption capacity of BN-coated samples tested in this study compared to other materials for three types of antibiotics (TET, CIP, and AMOX). It can be seen that BN coatings with a developed surface demonstrate a significantly higher efficiency of removing antibiotic molecules during adsorption treatment compared to other adsorbents. We did not find literature data on the adsorption of amphotericin B from aqueous solutions. In our case, the maximum sorption capacity is 269.7 mg/g. It is also worth noting that the adsorption capacity of the BN coating/substrate platform far exceeds that of spherical BN nanoparticles with smooth surface [36]: 502.78 mg/g versus 297.3 mg/g for TET and 315.75 mg/g versus 238.2 mg/g for CIP. This difference can be mainly explained by the larger specific surface area, which makes a decisive contribution in the case of an adsorption process. 

## 3. Materials and Methods

### 3.1. Preparation of BN Coatings 

Nano-structured BN coatings were obtained on a 1 × 1 cm^2^ silicon substrate in a horizontal tubular chemical vapor deposition reactor by a chemical interaction of amorphous boron particles with ammonia as described elsewhere [83].

### 3.2. Structural Characterization of BN Coatings

The coating morphology was analyzed using scanning electron microscopy (SEM) on a JSM-7600F (JEOL, Tokyo, Japan) instrument equipped with an energy dispersive X-ray (EDX) detector (Oxford Instruments, High Wycombe, UK). Chemical bonds after antibiotic absorption by BN coating were determined by Fourier-transformed infrared (FTIR) (Bruker Vertex 70V, Billerica, Massachusetts, USA) spectroscopy in the total reflection mode in the range of 400–4000 cm^−1^ with a resolution of 4 cm^−1^. X-ray diffraction (XRD) pattern of the coating was recorded on a D2 Phaser diffractometer (Bruker, Billerica, Massachusetts, USA) equipped with a position-sensitive detector (Elion) using the Bragg-Brentano geometry. The survey was carried out in the step-by-step scan mode in the 2θ range of 10°–80° at a scan step of 0.1° and an exposure time of 4 s.

### 3.3. Adsorption Studies 

Adsorption studies were performed using four types of antibiotics: tetracycline (Belmedpreparaty, Minsk, Belarus), ciprofloxacin (Dr. Reddy’s, Hyderabad, India), amoxicillin (Hemofarm A.D., Vrsac, Serbia) and amphotericin B (AMP) (OAO Sintez, Kurgan, Russia). Antibiotics were dry suspensions or coated tablets containing no impurities. The concentration was calculated, considering the amount of pure substance. Each type of antibiotics was completely dissolved in deionized water to prepare a stock solution, which was then diluted in pH 4, pH 7, and pH 9 buffer solutions to obtain working solutions with antibiotic concentrations of 10, 50, and 100 µg/mL. Then, one BN coating was added to 10 mL of each antibiotic solution. Adsorption tests were carried out at room temperature. Blank experiments were also carried out with an antibiotic, but without an adsorbent, and with an adsorbent, but without an antibiotic. 

Kinetic curves were plotted to evaluate the antibiotic removal efficiency. To do this, at certain time intervals (6 h, 1, 3, 5, 7, 11, 14, 21, and 28 days), 2 mL of the supernatant was taken and the residual concentrations of antibiotic solutions were determined by measuring absorbance values using a UV-visible spectrophotometer. For each type of antibiotic, calibration curves were preliminarily plotted based on measurements of the absorption intensity of antibiotic solutions in the concentration range of 0.5–4000 μg/mL. The antibiotic concentration in the solution at each measurement time point was determined from the calibration curve. All experiments were carried out in triplicate.

The antibiotic removal efficiency (*R*, %) was calculated using Equation (1), where *C*_0_ and *C*_t_ are the initial and antibiotic concentrations at time *t*, respectively (mg/L). The adsorption capacity (*q_e_*, mg/g) of BN coatings was studied after exposure for 48 days in an 1 mg/mL antibiotic solution and calculated using Equation (2) [15], where *C*_0_ and *C_e_* are the initial and equilibrium antibiotic concentrations, respectively (mg/L), *V* is the volume of the antibiotic solution (L), and *W* is the amount of adsorbent (g).
(1)$$R\left(\%\right)=\frac{\left(C_{0}-C_{t}\right)\times 100}{C_{0}},$$
(2)$$q_{e}=\frac{\left(C_{0}-C_{e}\right)\times V}{W}.$$

### 3.4. BN Coating Purification from Adsorbed Antibiotics 

Coatings were cleaned from antibiotics in acetonitrile solution, acetate buffer solution at pH 4.4 and ethanol as described elsewhere [36]. Studies were carried out for 28 days, after which desorption curves were plotted. The coating surface was analyzed by FTIR spectroscopy.

### 3.5. DFT Calculations

Theoretical modeling of the antibiotic sorption onto the BN surface was performed using density functional theory (DFT) [84,85] in the framework of the generalized gradient approximation (GGA) using normalized Trulier-Martins pseudopotentials [86] in the SIESTA software package [87]. The systems were modeled as a 7 × 4 BN unit cell. In order to smooth the intermolecular interactions in the non periodic direction, a sufficiently large vacuum gap was set in the *z* direction, so that the distance between the periodic structures was at least 15 Å. The plane wave energy cutoff was set at 200 Ry. To calculate the equilibrium atomic structures, the Brillouin zone was chosen according to the Monkhorst-Pack scheme [88] and was 4 × 4 × 1. Although the DFT method is widely used to calculate the electronic structure, it poorly describes the power of dispersion and van der Waals interactions, which can regulate the physical absorption process. Therefore, the Grimm correction method (DFT-D method) was used to model antibiotic-BN systems [89].

## 4. Conclusions

Here we show that nano-structured coatings of hexagonal boron nitride are promising sorbents for reusable wastewater treatment from various types of antibiotics. The maximum sorption capacity of tetracycline, ciprofloxacin amoxicillin, and amphotericin B was 502.78, 315.75, 400.17 and 269.7 µg/g, respectively. BN coatings show a significantly higher CIP, TET, and AMOX adsorption capacity compared to other adsorbents. As far as we are aware, the possibility of effective purification of water from amphotericin B has been showed for the first time.

The rate of removal of antibiotics on the first day was high in all cases of the experiment, then there is a gradual decrease in the rate of effective removal of antibiotics. Antibiotic removal efficiency can be represented as follows: TET > CIP > AMOX > AMP.

It is shown that the acidity of the medium significantly affects the kinetics of adsorption: with an increase in the medium’s acidity from pH 4, the efficiency increased; after pH 7 there was a gradual decrease in efficiency; however, it was higher in an alkaline environment than in an acidic one.

The possibility of repeated use of nano-structured hexagonal BN coatings is shown, which will positively affect its economic and environmental efficiency. Losing efficiency during repeated coating is no more than 5–15%, which confirms the prospects for reusable use.

Based on theoretical modeling data, it can be concluded that adsorption is the main process of water purification from antibiotics. The interaction of the antibiotics with the BN surface occurs through the OH group. In an acidic medium, the electron density redistribution is concentrated mainly on the antibiotic molecule and almost does not affect BN, which explains the weaker binding compared to an alkaline or neutral medium.

The obtained results clearly show the high potential of BN coatings as an affordable, economical and reusable adsorbent for effective water purification from antibiotics. An important result is the established dependence of sorption and desorption on the solution pH, which makes it possible to control the purification processes of solutions and adsorbents from antibiotics. In the future, it is important to study the selectivity of sorbents in the presence of various pollutants.

## Figures and Tables

**Figure 1 ijms-23-16097-f001:**
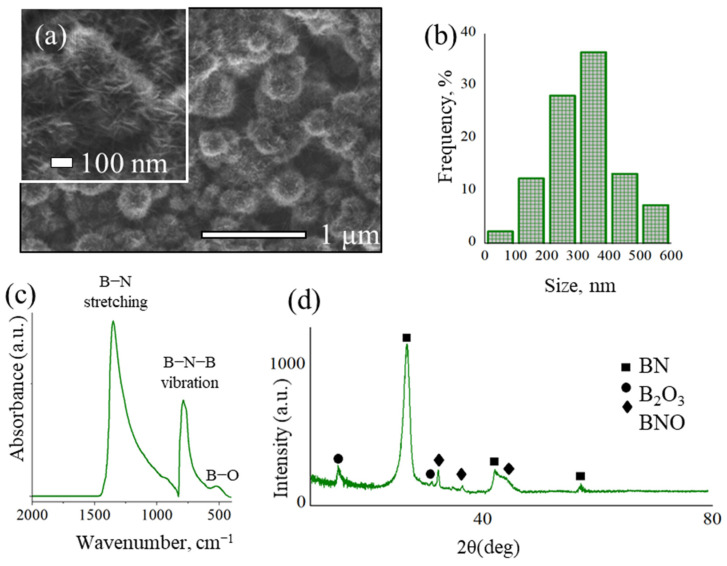
SEM micrographs at low and high (inset) magnifications (**a**), size distribution of BNNPs (**b**), FTIR spectrum (**c**) and XRD pattern (**d**) of BN coating.

**Figure 2 ijms-23-16097-f002:**
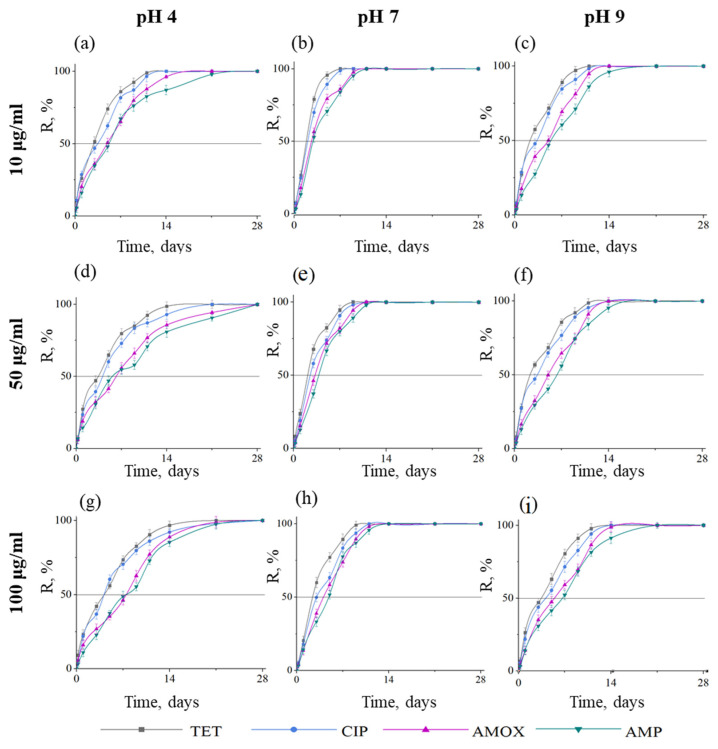
Kinetic curves showing the removal efficiency of four types of antibiotics with BN coated samples at pH 4 (**a**,**d**,**g**), pH 7 (**b**,**e**,**h**), and pH 9 (**c**,**f**,**i**) and initial antibiotic concentrations of 10 (**a**–**c**), 50 (**d**–**f**) and 100 µg/mL (**g**–**i**). T = 25 °C.

**Figure 3 ijms-23-16097-f003:**
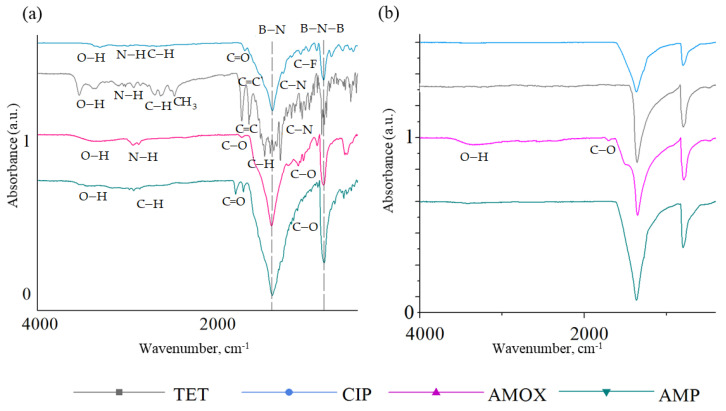
FTIR spectra of BN coatings before (**a**) and after (**b**) cleaning.

**Figure 4 ijms-23-16097-f004:**
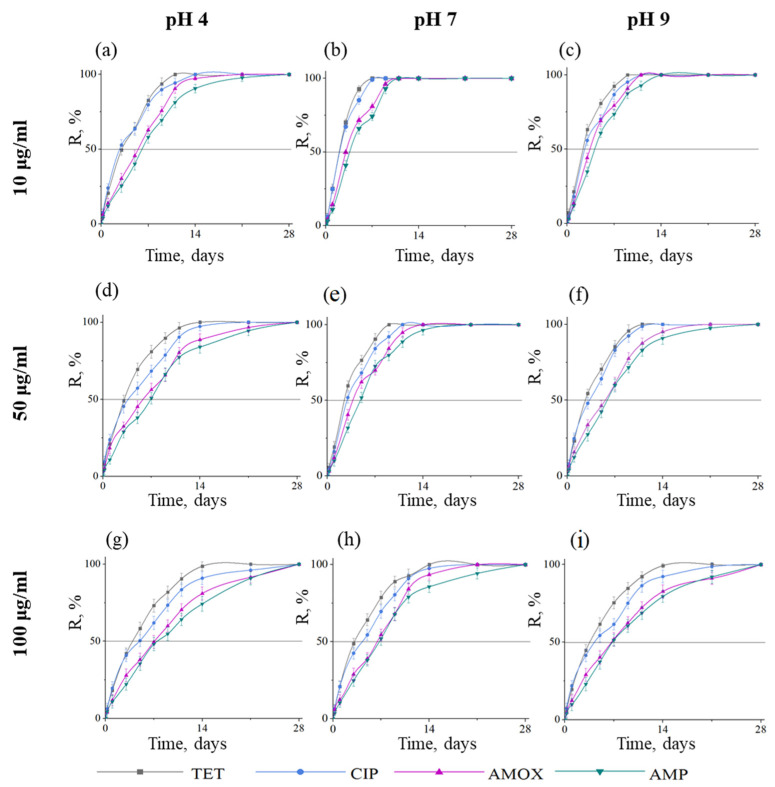
Kinetic curves showing the removal efficiency of four types of antibiotics during the second cycle (after cleaning of BN coated samples) at pH 4 (**a**,**d**,**g**), pH 7 (**b**,**e**,**h**), and pH 9 (**c**,**f**,**i**) and initial antibiotic concentrations of 10 (**a**–**c**), 50 (**d**–**f**) and 100 µg/mL (**g**–**i**). T = 25 °C.

**Figure 5 ijms-23-16097-f005:**
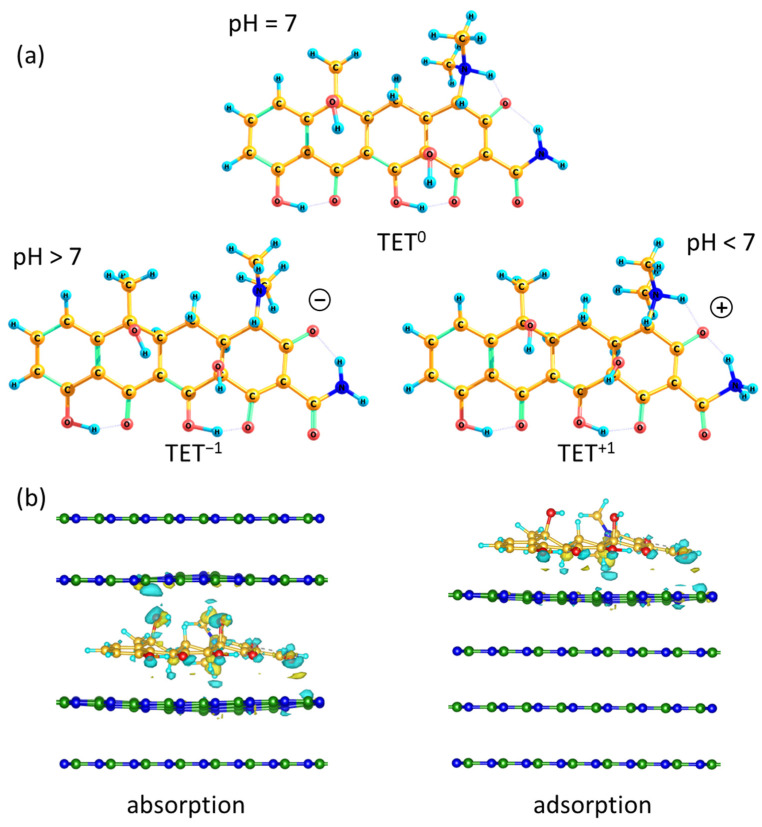
Schematics of tetracycline structure with different charges (**a**) and absorption and adsorption models of TET–BN systems (**b**). TET is considered in three states corresponding to three pH solutions: neutral (TET^0^), negatively (TET^−1^), and positively (TET^+1^) charged.

**Figure 6 ijms-23-16097-f006:**
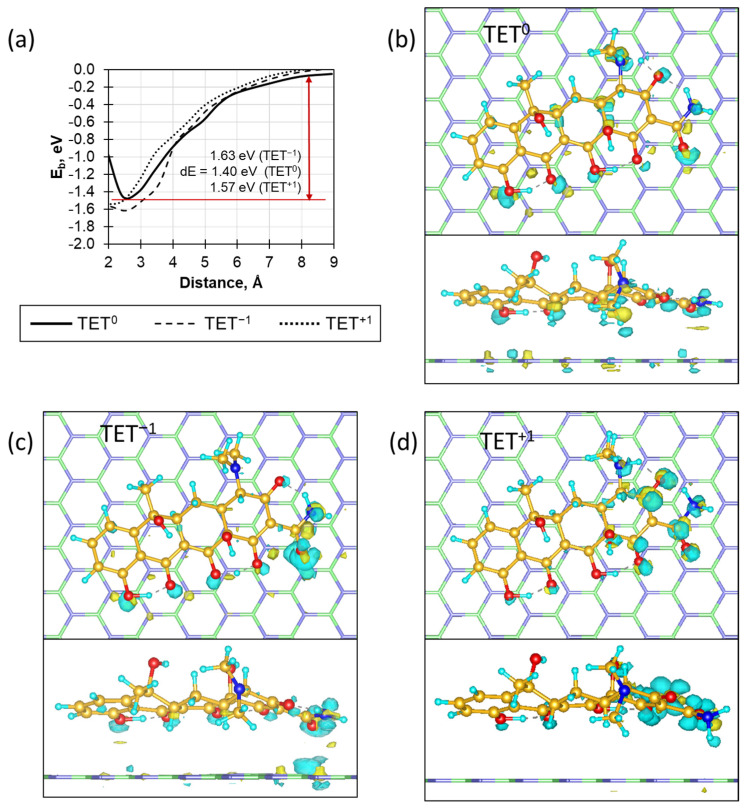
(**a**) The changes in binding energy as a function of the distance between tetracycline and the boron nitride surface for uncharged (solid line), negatively (dashed line) and positively (dotted line) charged forms of TET. The electron density redistribution at the BN/TET interface for (**b**) uncharged, (**c**) negatively and (**d**) positively charged forms of TET. The loss and gain of charge are denoted by yellowish and bluish colors, respectively. The boron, nitrogen, carbon, oxygen, and hydrogen atoms are marked by green, blue, yellow, red, and cyan colors, respectively. The isosurface constant value is 0.01 eV/Å.

**Table 1 ijms-23-16097-t001:** Antibiotic purification efficiency using BN coatings.

Antibiotic	Purification Efficiency, %	Initial Antibiotic Concentration, µg/mL	Time, Days		
			pH 4	pH 7	pH 9
TET	R_50_	10	2	2	2
		50	3	2	2
		100	4	2	3
	R_100_	10	12	7	11
		50	15	9	12
		100	18	10	14
CIP	R_50_	10	3	2	3
		50	4	3	3
		100	4	3	4
	R_100_	10	14	8	12
		50	21	10	14
		100	23	11	14
AMOX	R_50_	10	5	3	4
		50	5	3	5
		100	8	4	6
	R_100_	10	17	10	14
		50	28	11	14
		100	26	12	15
AMP	R_50_	10	5	3	5
		50	6	4	7
		100	8	5	6
	R_100_	10	23	11	18
		50	28	12	18
		100	28	14	21

**Table 2 ijms-23-16097-t002:** Sorption efficiency during first (I) and second (II) cleaning cycle at pH 7.

Time, Days	TET		CIP		AMOX		AMP	
	I-R,%	II-R,%	I-R,%	II-R,%	I-R,%	II-R,%	I-R,%	II-R,%
Initial concentration of antibiotic 10 µg/mL								
1	26.5	25.0	24.7	23.1	17.9	14.3	13.3	10.9
7	100	100	98.7	97.2	86.0	80.9	83.9	74.2
14	100	100	100	100	100	100	100	100
28	100	100	100	100	100	100	100	100
Initial concentration of antibiotic 50 µg/mL								
1	23.9	21.0	19.2	16.1	15.5	12.0	12.7	10.1
7	94.7	90.5	90.7	84.1	82.1	69.9	80.1	72.5
14	100	100	100	100	100	100	100	96.3
28	100	100	100	100	100	100	100	100
Initial concentration of antibiotic 100 µg/mL								
1	20.4	19.3	17.9	20.9	14.1	12.4	13.9	10.3
7	89.5	78.6	83.4	69.5	73.8	54.4	77.6	51.9
14	100	100	100	97.6	100	93.5	100	85.6
28	100	100	100	100	100	100	100	100

**Table 3 ijms-23-16097-t003:** Adsorption capacity of various materials in the purification of aqueous solutions from antibiotics.

Material	Adsorption Capacity (*q_e_*), mg/g	Material	Adsorption Capacity (*q_e_*), mg/g	Material	Adsorption Capacity (*q_e_*), mg/g
Tetracycline		Ciprofloxacin		Amoxicillin	
Graphene oxide/calcium alginate composite fibers [70]	131.6	Powdered activated carbon magnetized by iron(III) oxide NPs [71]	109.6	Zinc oxide coated carbon nanofiber composite [72]	156.0
Graphene oxide [47]	313	Activated carbon magnetized with iron (III) oxide NPs [48]	178.7	Mn-impregnated activated carbons [73]	122.0–132.0
Fe-doped zeolite [74]	200.0	Chalcogenide based magnetic adsorbent [75]	181.3	Magnetic multi-walled carbon nanotubes [76]	50.0
Shrimp shell waste [77]	230.0	Hydrogel derived from agrowaste [78]	106.9	Quaternized cellulose from flax noil [79]	183.1
Iron(III)-loaded cellulose nanofibers [80]	294.1	Amine-functionalized MCM-41 mesoporous silica NPs [81]	164.3	Magnetic adsorbent [82]	238.1
BN coatings *	502.8	BN coatings *	315.4	BN coatings *	400.2

* this study.

## Data Availability

Not applicable.

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
