# Peer review of "Efficient and Reusable Sorbents Based on Nanostructured BN Coatings for Water Treatment from Antibiotics"

_ijms, 2022, doi:10.3390/ijms232416097_

Round 1

Reviewer 1 Report

The authors presented their work in a very lucid and scientific manner and could be easy for the readers to grasp the intent of the work. The work can help the readers and workers working in this area to find new results. However, there are a few comments and suggestions for your kind notice.

Author Response

Reviewer 1

We thank the reviewer for helpful comments and valuable suggestions on the structure and content of our manuscript. We revised the manuscript accordingly, indicated all the changes in the text made in responses to the reviewers. The detailed corrections are listed below in a point-by-point style.

  1. Mention the other ways which bring antibiotic contamination in water (line 34-35)

Additional information were added in manuscript as follows (lines 36-39):

In addition, extremely high concentrations of antibiotics have been reported in wastewater from antibiotic production plants (oxytetracycline: 32.0 mg/L), livestock farms (oxytetracycline: 2.1 mg/L), hospitals (ciprofloxacin (CIP): 0.9 mg/L), urban environment (CIP: 0.25 mg/L), and near aquaculture farms (sulfamethoxazole: 5.6 mg/L) [7–10].

  1. 2D material properties mentioned can be brought to the starting of the paragraph.

We structured the introduction in such a way that we first discuss antibiotics and the problems that arise when using them, then we consider methods for removing antibiotics, and then we move on to specific materials - 2D sorbents and boron nitride. We consider this presentation of the material quite logical. However, in response to other questions from reviewers, the introduction has been supplemented with additional information.

  1. R50 value increase with the increase in the initial concentration of antibiotics (line 116) is not observed in table 1

The change in R50 depending on the concentration is described in the text as follows, see lines 146-151, “In an acidic environment (pH 4), R50 is reached after 2, 3 and 4 days (TET), 3, 4 and 4 days (CIP), 5, 5 and 8 days (AMOX) and 5, 6 and 8 days (AMP), respectively, at initial concentrations of 10, 50 and 100 µg/mL. In a neutral environment (pH 7), purification is faster. At initial concentrations of 10, 50 and 100 µg/mL, the R50 values are 2, 2, and 2 days (TET), 2, 3, and 3 days (CIP), 3, 3, and 4 days (AMOX), and 3, 4, and 5 days (AMP), respectively.”

Note that for ease of understanding, the R50 values have been rounded to days. With this approximation, for TET at pH 7, the R50 values at different concentrations are the same (2, 2, and 2 days for 10, 50, and 100 µg/mL, respectively). However, as can be seen from Figure 2, the R50 values are 1 day 19 h, 2 days 2 h and 2 days 7 h at these concentrations. However, we believe that such accuracy only complicates the perception of the results.

  1. The adsorption efficiency variation with pH for AMP is not mentioned

Additional information were added in manuscript as follows (lines 187-194):

The AMP charge depends on pH and leads to different adsorption activity. In our case, with an increase in pH from 4 to 7, the adsorption properties increase, but in an al-kaline medium they decrease again. This can be explained by the fact that the zeta poten-tial increases from -34.5 to -26 mV with an increase in pH from 4 to 7, this is due to the dissociation of the carboxylate moiety of AMP [65]. A decrease in the rate of AMP removal with a further increase in pH to 9 may be because of electrostatic repulsion from the BN surface, since both the carboxyl and amino groups are deprotonated under alkaline condi-tions, which leads to more negatively charged AMP.

  1. In section 2.3, some typos like CH3, cm-1, etc are present which should be omitted.

Thanks for the comment, these typos were corrected (section 2.3, starting at line 200)

  1. write full form of short form used initially like “BE, CVD” etc. (line 247, 291)

Thanks for the comment; we have given a transcript of the abbreviations (line 281, 282, 321)

  1. In conclusion (line 357) TC should be replaced by TET

Done (line 393)

Reviewer 2 Report

In this manuscript, the author focused on the removal of several antibiotics using a nanostructured h-BN adsorbent. The characterization aspect of the prepared composites was sufficient, and the practical aspect was reasonably and sufficiently studied. Hence, I recommended a major review and it is accepted for this journal after the author clarifies the following comments.

1.      The authors may need to briefly address the difference(s) between the current manuscript and other similar published articles in the Introduction section.

2.      The authors should highlight the importance and novelty of the developed system.

3.      On what basis the authors chose these particular antibiotics? Are these pollutants more abundant in wastewater as compared to others? Are they more toxic?

4.      In adsorption studies, surface area plays an important role, the author should present the BET-specific surface areas of the adsorbents that are used in this work.

5.      Did the authors measured the surface charge of the adsorbent (zeta potential)? For instance, I suggest the authors to measure zeta potential values of the materials as function of pH.

6.       Which type of water was used to prepare the solutions with the antibiotics: ultra-pure water, distilled water or deionized water? This detail is important for the readers.

7.      The biggest concern with the approach presented here is the potential for sorption of unintended species in the water. Reasonably capacity is demonstrated, which is important, but selectivity is at least as important as capacity. The authors show that the sorbent can capture the antibiotics, but this study does not evaluate how many interfering species are also captured, which will limit the practical effectiveness. This topic should be addressed explicitly in the discussion.

8.      Did the authors check any interferences in spectrophotometer measurements of dye from the matrix of the synthesized materials? Please comment.

9.      I suggest the authors to discuss briefly the scale-up of the process (problems, possibilities), and the economy and/or energy efficiency of the process.

10.   Conclusions need to be improved by specifying the discussed important points within this work. In the conclusions, the authors should also provide an outlook of the challenges and potential future directions.

Author Response

We thank the reviewer for helpful comments and valuable suggestions on the structure and content of our manuscript. We revised the manuscript accordingly, indicated all the changes in the text made in responses to the reviewers. The detailed corrections are listed below in a point-by-point style.

  1. The authors may need to briefly address the difference(s) between the current manuscript and other similar published articles in the Introduction section.
  2. The authors should highlight the importance and novelty of the developed system.

Additional information were added in manuscript as follows (lines 60-63):

“These methods have a number of disadvantages, such as introduction of active organisms into the aquatic environment during water treatment, low selectivity [33–35], rapid anni-hilation of photogenerated carriers, and incomplete mineralization [32]”

and line 88-92:

Despite significant progress in wastewater treatment from antibiotics, the relevance of developing highly effective and safe adsorbents remains high. The development of adsorbents based on hexagonal boron nitride will make it possible to obtain an affordable and cost-effective reusable adsorbent for more efficient water purification from antibiotics. In addition, sorption to amphotericin B was studied for the first time.

  1. On what basis the authors chose these particular antibiotics? Are these pollutants more abundant in wastewater as compared to others? Are they more toxic?

Additional information were added in manuscript as follows, from line 49:

Amoxicillin, ciprofloxacin, and tetracycline are among the classes of most commonly prescribed antibiotics [17–20]. Tetracycline and ciprofloxacin are major wastewater pollutants in East and Southeast Asia [21]. According to the World Health Organization report for 2015-2016, amoxicillin is one of the most used antibiotics in the world [22] and therefore is often detected in wastewater [23,24]. Amphotericin B is a common antifungal agent that has been used for more than half a century [25,26,27], traces of which are also often found in the environment [28]. Several microorganisms, such as Aspergillus terreus, were reported to develop resistance to amphotericin B [29], which requires its removal from wastewater.

  1. In adsorption studies, surface area plays an important role, the author should present the BET-specific surface areas of the adsorbents that are used in this work.

Additional information were added in manuscript as follows, from line 118:

The specific surface area of BN coating was measured by low-temperature nitrogen adsorption on a NOVA 1200e instrument (Quantachrome, USA). The obtained results were processed using the Brunauer-Emmett-Teller (BET) equation. Prior to adsorption measurements, the samples were degassed in vacuum at 200 °C overnight. The surface area of the nanostructured BN coating was 90.61 m2/g.

  1. Did the authors measured the surface charge of the adsorbent (zeta potential)? For instance, I suggest the authors to measure zeta potential values of the materials as function of pH.

Additional information were added in manuscript as follows, from line 123:

The zeta potential of BN was determined using a Zetasizer Nano-ZS ZEN3600 in-strument (Malvern). The charge of the synthesized pure BN coating at pH 7 is -26 mV. It was reported that a change in the acidity of the medium did not affect the BN surface charge [64], therefore; the BN charge was considered unchanged.

  1. Which type of water was used to prepare the solutions with the antibiotics: ultra-pure water, distilled water or deionized water? This detail is important for the readers.

The paper states that deionized water was used, see line 335

  1. The biggest concern with the approach presented here is the potential for sorption of unintended species in the water. Reasonably capacity is demonstrated, which is important, but selectivity is at least as important as capacity. The authors show that the sorbent can capture the antibiotics, but this study does not evaluate how many interfering species are also captured, which will limit the practical effectiveness. This topic should be addressed explicitly in the discussion.

We agree that selectivity is also an important characteristic of adsorbents; however, at this stage, we studied only the effect of antibiotic concentration and medium acidity. Finally, we added a comment regarding future research.

  1. Did the authors check any interferences in spectrophotometer measurements of dye from the matrix of the synthesized materials? Please comment.

Additional comment was added, line 338: “Blank experiments were also carried out with an antibiotic, but without an adsorbent, and with an adsorbent, but without an antibiotic.”

  1. I suggest the authors to discuss briefly the scale-up of the process (problems, possibilities), and the economy and/or energy efficiency of the process.

It is quite difficult to estimate the cost of adsorbents obtained under laboratory conditions. However, the proposed method for producing BN nanoparticles is quite simple and easily scaled up, so we added a comment that it could be cost effective (line 400).

  1. Conclusions need to be improved by specifying the discussed important points within this work. In the conclusions, the authors should also provide an outlook of the challenges and potential future directions.

The conclusion was rewritten, see from line 376.

Round 2

Reviewer 2 Report

Authors have revised the manuscript according the recommendations, and answered the questioned points. Now it looks suitable for publication.